# TNBC Therapeutics Based on Combination of Fusarochromanone with EGFR Inhibitors

**DOI:** 10.3390/biomedicines10112906

**Published:** 2022-11-12

**Authors:** Natalie Carroll, Reneau Youngblood, Alena Smith, Ana-Maria Dragoi, Brian A. Salvatore, Elahe Mahdavian

**Affiliations:** 1Jackson Laboratory, Sacramento, CA 95838, USA; 2Feist-Weiller Cancer Center, LSU Health-Shreveport, Shreveport, LA 71104, USA; 3Department of Chemistry and Physics, LSU Shreveport, Shreveport, LA 71115, USA; 4Department of Biological Sciences, LSU Shreveport, Shreveport, LA 71115, USA

**Keywords:** TNBC, fusarochromanone, EGFR, mTOR, everolimus, erlotinib, lapatinib, drug combinations, synergistic, phenotypic screens

## Abstract

Fusarochromanone is an experimental drug with unique and potent anti-cancer activity. Current cancer therapies often incorporate a combination of drugs to increase efficacy and decrease the development of drug resistance. In this study, we used drug combinations and cellular phenotypic screens to address important questions about FC101′s mode of action and its potential therapeutic synergies in triple negative breast cancer (TNBC). We hypothesized that FC101′s activity against TNBC is similar to the mTOR inhibitor, everolimus, because FC101 downregulates the phosphorylation of two mTOR substrates, S6K and S6. Since everolimus synergistically enhances the anti-cancer activities of two known EGFR inhibitors (erlotinib or lapatinib) in TNBC, we performed analogous studies with FC101. Phenotypic cellular assays helped assess whether FC101 acts similarly to everolimus, in both single and combination treatments with the two inhibitors. FC101 outperformed all other single treatments in both cell proliferation and viability assays. However, unlike everolimus, FC101 produced a sustained decrease in cell viability in drug washout studies. None of the other drugs were able to maintain comparable effects upon removal. Although we observed slightly additive effects when the TNBC cells were treated with FC101 and the two EGFR inhibitors, those effects were not truly synergistic in the manner displayed with everolimus.

## 1. Introduction

### 1.1. Cancer Statistics and Treatment Modalities

Cancer continues to be a significant human health challenge, claiming 10 million lives worldwide each year [1]. In 2022, an estimated 1.9 million new cancer cases will be diagnosed in the United States, and over 609,360 cancer deaths will occur [2]. Although surgery, radiation, and therapeutics are often effective against cancer when the disease is detected early, many cancer cases are often only diagnosed after the primary tumor cells have metastasized [3]. Systemic chemotherapy, which uses nonselective drugs that affect both cancer cells along with rapidly growing normal cells, causes severe side effects and preventing effective long-term use [4]. Improved clinical efficacy and safety have been achieved with molecularly targeted cancer therapies, which represent a paradigm shift in oncology [4]. To improve efficacy and safety in cancer treatment, molecularly targeted cancer therapies have emerged, representing a paradigm shift in oncology [5]. Targeted therapies are tailored to disrupt a specific molecular entity that drives cancer development and progression [6]. These targets are part of a signaling cascade that has run awry, and their modulation can reverse the phenotypic cancerous progression. Unfortunately, the clinical outcomes of targeted drugs used as single treatments for advanced cancer have been disappointing [7]. Genomic sequencing has revealed that advanced cancer acquires complex genomic heterogeneity to support its abnormal phenotypes [8]. This heterogeneity is fueled by genome instability that is encoded at the functional proteomic level, including oncogenic and non-oncogenic signaling networks [8]. Furthermore, advanced cancer evolves multiple escape mechanisms to evade inhibition of the primary drug target. This reprograming activates cancer cell survival mechanisms, leading to drug resistance and poor clinical outcomes [9]. Drug resistance represents a significant clinical challenge that requires targeting any compensatory pathways that develop to support the drug-resistant state [10]. Advances in cancer genome sequencing have provided a reliable experimental framework to identify both the molecular drivers of drug response and drug resistance [11]. Dual cancer therapeutics targeting both the primary oncogenic and the secondary adaptive survival pathways have the potential to significantly improve clinical outcomes [10,11]. Thus, rational combination therapies have emerged to overcome drug resistance, representing a second paradigm shift in cancer treatment. Preclinical investigation of the synergy of combination therapies for advanced cancer is an area of increasing interest in oncology research [11,12].

### 1.2. Triple Negative Breast Cancer (TNBC)

Breast cancer is the most common malignancy in women and the second most common cancer worldwide [1,13]. It is highly heterogeneous, with subtypes displaying distinct genomic, proteomic, morphologic, and clinical behavior [1]. Triple positive breast cancer (TPBC) is characterized by the overexpression of three types of membrane protein receptors: HER^+^, ER^+^, and PR^+^ [1]. While effective targeted therapeutics against TPBC do exist, including endocrine therapy (tamoxifen) and anti-HER therapies (herceptin, lapatinib), there are no effective targeted therapeutics for TNBC, which lacks the prominent expression of HER2, ER, and PR receptors [1]. A large body of experimental evidence has established the importance of EGFR (HER1) receptor overexpression, hence its significance as a therapeutic target in TNBC [14,15]. However, targeting EGFR in TNBC using single-action kinase inhibitors (lapatinib and erlotinib) leads to drug resistance and poor clinical outcomes. The cancer cells compensates by unleashing inhibitory feedback on EGFR signaling [16]. This activates mTOR and PI3K-AKT, two major signaling pathways downstream from EGFR [16]. Thus, the disruption of cooperation and crosstalk between EGFR and either mTOR or PI3K-AKT signaling has received significant attention in efforts to develop effective treatments for TNBC [16]. Targeting mTOR activation to overcome the EGFR-directed drug resistant phenotype in TNBC has been well documented in recent research [16,17]. Preclinical data have indicated a significant improvement and synergy in the anti-cancer drug response in TNBC when EGFR and mTOR inhibitors are combined compared with single therapeutic approaches [18,19]. Notably, the combination treatment consisting of erlotinib and lapatinib (EGFR inhibitors) and everolimus (mTOR inhibitor) has advanced to clinical trials for the treatment of TNBC [19]. There is currently an increasing demand for other novel mTOR inhibitors to establish and validate new drug combinations that can effectively block the oncogenic networks responsible for drug resistance in TNBC.

### 1.3. The Potential of Fusarochromanone as an mTOR Inhibitor

In this context, we believe that the small molecule amino flavonoid fusarochromanone (FC101) possesses remarkable potential as an mTOR pathway modulating agent for use in cancer drug discovery. Lee et al. first isolated and determined the structure of FC101 via NMR and mass spectrometry (Figure 1) [20]. The unique therapeutic potential of FC101 merited the issuance of three patents covering its total synthesis along with its use as a treatment for solid tumors and angiogenic diseases [21]. Salvatore et al. synthesized FC101 in enantiomerically pure form. This has allowed further investigation of its therapeutic potential and provided new opportunities to modify the molecule for its clinical development. FC101 is a cancer-specific cytotoxic agent that exhibits 10- to 100-fold greater effects against cancer cells than against normal cells of the same type. This is the result of the differential uptake/permeability by cancer cells and selective toxicity towards them [21]. Furthermore, the most invasive cancer cells, particularly the oncogenic BRAF-mutant/MAPK-driven cancers, are the most sensitive to FC101′s anti-cancer activity [22]. This includes significant in vitro inhibition of the proliferation (sub-μM IC_50_) and migration, as well as the induction of apoptosis in *in vitro* models of melanoma, bladder cancer, and TNBC [22,23].

While FC101′s exact mechanism of action remains unknown, its mode of action may be similar to those of the inhibitors of the mTOR and MAPK signaling pathways. FC101 modulates the MAPK and mTOR pathways downstream of EGFR, both of which are vital to cancer cell proliferation, survival, and development of drug resistance in advanced cancer (e.g., TNBC). Prior Western blotting experiments confirmed that FC101 simultaneously inhibits the activity of the MAPK pathway (corresponding to p-ERK reduction) and the mTOR pathway (corresponding to p-S6K and p-S6 reduction) in MAPK-driven *TNBC* cancer cells (Appendix A). Thus, the multifaceted effects of FC101 on mTOR/MAPK in TNBC warrant further investigation of its potential for a synergistic anti-cancer drug response. For this reason, we focused on FC101′s potential as an mTOR inhibitor to overcome the single-therapy-driven drug resistance produced by known EGFR inhibitors. Specifically, the purpose of this project was to evaluate FC101′s synergistic activity in combination with two known EGFR inhibitors (erlotinib and lapatinib) through cellular phenotypic screens in TNBC. We used the known mTOR inhibitor everolimus as a control drug to provide a benchmark for FC101′s anticancer activity in both single and combination treatments. This study will aid in the elucidation of FC101′s mode of action.

### 1.4. Drug Panel

Our drug panel included FC101, erlotinib, lapatinib, and everolimus. The selection criteria for this group of kinase inhibitors incorporated experimental data on selectivity, potency, and clinical relevance. We chose FDA-approved kinase inhibitors with proven safety profiles that were the most selective and potent for their respective targets. Both erlotinib and lapatinib are small molecule synthetic drugs approved by the FDA for the treatment of human metastatic cancer [24]. Both drugs belong to a class of reversible EGFR kinase inhibitors, but they differ in structure, target specificity, and molecular mechanism of action. Erlotinib binds to EGFR’s ATP-binding site and is a type-I competitive inhibitor, whereas lapatinib binds to EGFR’s allosteric site and is a type-II allosteric inhibitor [25]. Structurally, erlotinib has a smaller head group, consisting of a substituted quinazolinamine ring, compared to lapatinib’s larger quinazoline head group [26]. In 2004, erlotinib was approved for use by the FDA as a single drug in the treatment of patients with metastatic non-small cell lung cancer and pancreatic cancer [27]. Lapatinib was approved by the FDA in 2007 for the treatment of advanced metastatic breast cancer in conjunction with the chemotherapy drug capecitabine [28]. Although both lapatinib and erlotinib inhibit the heterodimerization of EGFR and HER2, their signal transduction effects induced on downstream kinases differ slightly [29]. Lapatinib prevents the activation (and thus the phosphorylation) of both ERK1/2 and AKT [29], whereas erlotinib blocks the activation and thus the phosphorylation of ERK1/2 only [25].

The selective mTOR inhibitor, Everolimus is a structural analog of rapamycin but has slightly different drug-like properties [30]. Like rapamycin, everolimus does not directly inhibit the catalytic activity of mTOR, but instead it binds to the intracellular protein, FKBP12, promoting its interaction with the mTOR complex at a non-catalytic domain (adjacent to the kinase domain). This bimolecular association causes the release of raptor (a regulatory-associated protein of mTOR) from the newly formed drug/mTOR complex. Without raptor, mTOR1 is rendered catalytically inactive towards the phosphorylation of its downstream proteins, 4E-BP1/S6k. Phosphorylation of 4E-BP1/S6k is required for its interaction with eukaryotic initiation factors (eIF4B, eIF4E), which regulate the activation of protein synthesis. Thus, everolimus effectively downregulates protein synthesis that is vital for the cellular proliferation and survival of cancer [30]. 

Our previous experiments indicated that FC101′s anti-cancer activity may be directly related to its inhibition of the mTOR signaling pathway. We explored this hypothesis and evaluated FC101′s potential to produce synergistic anti-cancer effects in combination treatment modalities in TNBC using three cellular phenotypic screens. We screened drug responses in TNBC that resulted from either single or combination treatments with FC101, two EGFR inhibitors (erlotinib and lapatinib), and a known mTOR inhibitor (everolimus) as a control. After establishing the optimal doses from single-treatment experiments, we paired FC101 with either of the EGFR inhibitors and compared the drug response to that of everolimus. We used three well-documented phenotypic assays, growth (crystal violet assay), viability (MTT assay), and survival (drug washout assay).

## 2. Materials and Methods

### 2.1. Drug Treatments

The drug panel of kinase inhibitors for this project included FC101, erlotinib, lapatinib, and everolimus. Since FC101 is not commercially available in non-racemic form, it was synthesized in our laboratory in the Department of Chemistry (LSUS). Lapatinib (Tykerb), erlotinib (Tarceva), and everolimus (Afinitor) were purchased from Selleckchem, Inc. (Houston, TX, USA). Three cellular phenotypic screens (crystal violet, MTT, and drug washout) were used to determine the overall drug response of each compound in single and combination treatment modalities in TNBC (MDA-MB-231).

### 2.2. Cell Lines and Mammalian Cell Culture

The triple negative breast cancer (TNBC) cell line, MDA-MB-231, was provided by Dr. Ana-Maria Dragoi at the LSUHSC-INLET lab (Shreveport, LA, USA). The cells were grown in a culture medium of high glucose Dulbecco’s modified Eagle’s medium (DMEM, 02-0111-0500). All media were supplemented with 10% fetal bovine serum (FBS, S11550, Sigma Aldrich, St. Louis, MO, USA) and 1% penicillin/streptomycin (P4333, Sigma Aldrich). Trypsin was purchased from Sigma Aldrich (25-0510). Cells were incubated in appropriate culture plates or microwell plates and incubated in a humidified incubator at 37 °C and 5% CO_2_.

### 2.3. Crystal Violet Staining Assay

Cells were plated at a density of 10,000 cells per 100 μL of medium in a 96-well tissue culture plate. A dose–response relationship was established for each drug using drug concentrations of 0.05 μM, 0.1 μM, 0.5 μM, 1.0 μM, 1.5 μM, 2 μM, and 3 μM in DMSO. Cells were placed in a humidified incubator at 37 °C and 5% CO_2_. After 48 h of treatment, the growth medium was aspirated from the wells. The cells were gently washed twice with water (150 μL). Cells were fixed with 100% methanol (100 μL) for 20 min. The methanol was aspirated and crystal violet staining solution (40 μL) was added to each well and allowed to incubate at room temperature for 1 h. The staining solution was removed by aspiration and the wells were rinsed 3 times with water (250 μL). The plates were inverted and gently tapped on a paper towel to remove excess moisture and then dried overnight. The next day, 10% aq. acetic acid (50 μL) was added to each well and allowed to incubate for 2 h. A second set of crystal violet staining experiments was performed after determining a set of optimal doses for each drug based on the dose–response curves and the MTT viability assay. The test samples consisted of single treatments as well as combination treatments. Single treatments included 0.5 μM FC101, erlotinib, and everolimus, and 1 μM lapatinib in DMSO. Combination treatments included 0.5 μM FC101 with 0.5 μM erlotinib or 1 μM lapatinib as well as 0.5 μM everolimus with 0.5 μM erlotinib or 1 μM lapatinib in DMSO. In addition, DMSO-treated negative control wells were included. Cells were placed in a humidified incubator at 37 °C and 5% CO_2_ for 24, 48, or 72 h. The absorbance in each well was read at 595 nm using a microplate reader SpectroMax M2/M2e spectrometer (Molecular Devices, San Jose, CA, USA).

### 2.4. MTT Viability Assay

A Trevigen TACS MTT Cell Assay (Cat # 4890-25-K) was used to measure cell viability. Cells were plated at a density of 10,000 cells per 100 μL medium in a 96-well tissue culture plate. The next day the growth medium was aspirated and fresh growth media with the respective treatment conditions were added. The test samples consisted of single and combination drug treatments. Single treatments included 0.5 μM FC101, erlotinib, and everolimus, and 1 μM lapatinib. Combination treatments included 0.5 μM FC101 with 0.5 μM erlotinib or 1 μM lapatinib as well as 0.5 μM everolimus with 0.5 μM erlotinib or 1 μM lapatinib. In addition, DMSO-treated negative control wells were included. Cells were placed in a humidified incubator at 37 °C and 5% CO_2_ for 24, 48, or 72 h. At the appropriate time point, 10 μL of the MTT reagent (5 mg/mL) was added to each well, and the plates were returned to the incubator for 2 h until the purple dye was visible. Once the dye was visible, detergent reagent (100 μL) was added to each well, and the plates were immediately covered with foil. Plates were placed in a dark drawer to incubate overnight at room temperature. The absorbance in each well was read at 570 nm using a microplate reader SpectroMax M2/M2e spectrometer (Molecular Devices, San Jose, CA, USA).

### 2.5. Drug Washout Assay

To determine whether the effects of the drug treatments would still have an impact after the drugs were removed, we performed a drug washout assay. The drugs were incubated with the cells for a total of 72 h and then the growth medium was removed, and fresh drug-free growth medium was added. The cells were then allowed to recover for another 72 h. The cells were transferred to a 96-well microtiter plate (10,000 cells per 100 μL; six replicates). Cell viability was measured at 72 h following drug treatment and again 72 h after drug washout using a Trevigen TACS MTT Cell Assay. Treatment concentrations included FC101 (0.5 μM), erlotinib (0.5 μM), lapatinib (1 μM), everolimus (0.5 μM) in DMSO, with the same concentrations used for combinations of FC101 or everolimus with erlotinib and lapatinib.

### 2.6. Statistical Analysis

All cell culture experiments including crystal violet, MTT, and drug washout assays were performed three times in triplicate. All graphing and statistical analyses were performed in GraphPad Prism. Statistical significance was determined using an ANOVA. *p*-values less than 0.01 were considered statistically significant. Results are reported as mean ± SEM (n = 3), *p* < 0.01.

## 3. Results

### 3.1. Crystal Violet Proliferation Assay

For all single drug treatments, a dose range of 0.05–3.0 μM was used to treat MDA-MB-231 cancer cells in a 96-well cell culture plate for 48 h. The dose–response curve shown in Figure 2 represents the normalized color intensity (% control, A_595_) versus the drug concentration for each experiment. For dual drug treatment experiments, a concentration of 0.5 μM was chosen for FC101, erlotinib, and everolimus, and a concentration of 1.0 μM was chosen for lapatinib.

A second round of crystal violet assays was then conducted with each chosen dose combination at time points of 24, 48, and 72 h (Figure 3). At 48 h, all single and combination treatments had significantly lower proliferation levels than the untreated control cells (*p* < 0.01) (Figure 3).

In single drug treatment experiments, FC101 (0.5 μM) induced the greatest inhibition of cell proliferation compared to lapatinib (1.0 μM), erlotinib (0.5 μM), or everolimus (0.5 μM). The combination treatment of FC101 (0.5 μM) and lapatinib (1.0 μM) had a slightly lower average A_570_ value, but there was no significant difference between the two when compared with FC101 (0.5 μM) treatment alone. Similarly, the combination treatment of FC101 (0.5 μM) and erlotinib (0.5 μM) had a slightly lower average A_570_ value, but there was also no significant difference between the two when compared with FC101 (0.5 μM) treatment alone. The combination of everolimus (0.5 μM) and lapatinib (1.0 μM) had a significantly lower level of proliferation than treatment with everolimus (0.5 μM) alone. However, in combination of everolimus (0.5 μM) and erlotinib (0.5 μM), proliferation was not significantly different than that of treatment with everolimus (0.5 μM) alone.

### 3.2. MTT Viability Assay

Inhibition of cellular viability is a crucial feature of any anti-cancer agent against TNBC. The MTT assay measures cellular viability by evaluating the mitochondrial integrity and activity of dehydrogenases. Metabolically active and viable cells have intact mitochondria and therefore have high dehydrogenase enzyme activity. Most anti-cancer drugs induce cytotoxicity by compromising the mitochondrial integrity; thus, the reduction in dehydrogenase enzyme activity can be used as an end point for drug response on viability.

Single and combination treatments of FC101 and lapatinib at various concentrations were first tested in a 48-h MTT assay as shown in Figure 4. In single treatment experiments, FC101 (0.05 μM and 0.1 μM) did not induce a significant reduction in TNBC cellular viability compared to control treatment with DMSO (*p* < 0.001). However, the viability of FC101-treated cells at 0.5 μM was significantly lower than that of the DMSO-treated control cells (*p* < 0.001). All the single treatments with lapatinib (0.1 μM, 0.5 μM, and 1.0 μM) displayed significantly lower cell viability levels than the control cells. When analyzing the combination of FC101 and lapatinib against the respective equivalent concentration of FC101 alone, the only combination treatment that displayed significantly lower viability was the combination of FC101 (0.05 μM) and lapatinib (1.0 μM) (*p* < 0.01). However, when the same combination was analyzed against the equivalent single lapatinib dose of 1.0 μM, the significant difference disappeared, indicating that the effect was due to lapatinib rather than a synergistic effect. Except for the combination of FC101 (0.05 μM) and lapatinib (0.1 μM), which did not display significantly lower viability than control cells, cell viability for all the other FC101 and lapatinib combination treatments was significantly lower than that of the control cells. From these results, the combination of 0.5 μM FC101 and 1.0 μM lapatinib was selected for use in future time response experiments because it was the most effective drug dose and had significantly (*p* < 0.001) lower viability when compared to control cells (Figure 4). MTT time response curve results for the combination treatments of FC101 and everolimus with each of the EGFR inhibitors (lapatinib and erlotinib) at the time points of 24, 48, and 72 h are shown in Figure 5.

At 48 h, the cell viability in all single and combination treatments was significantly lower than in the control experiments (*p* < 0.001) (Figure 6). Viability for the combination of FC101 (0.5 μM) and lapatinib (1.0 μM) was significantly lower than FC101 (0.5 μM) alone or lapatinib (1.0 μM) alone (*p* < 0.001). Viability for the combination of FC101 (0.5 μM) and erlotinib (0.5 μM) was also significantly lower than FC101 (0.5 μM) alone or erlotinib (0.5 μM) alone (*p* < 0.001). Viability for the combination of everolimus (0.5 μM) and lapatinib (1.0 μM) was significantly lower than everolimus (0.5 μM) alone or lapatinib (1.0 μM) alone (*p* < 0.001). Viability for the combination of everolimus (0.5 μM) and erlotinib (0.5 μM) was also significantly lower than everolimus (0.5 μM) alone or erlotinib (0.5 μM) alone (*p* < 0.001). When comparing the FC101-based combinations with the everolimus-based combinations, there was no significant difference in viability between the combination of FC101 with erlotinib versus the combination of everolimus and erlotinib. However, we observed a different result when comparing the viability of the FC101/lapatinib combination to that of the everolimus/lapatinib combination; the everolimus/lapatinib combination had a significantly lower viability than the FC101/lapatinib combination. Viability for treatment with FC101 (0.5 μM) alone was significantly lower than all single treatments with lapatinib, erlotinib, or everolimus. Treatment with FC101 alone did not show any significant difference when compared to the combination treatment of everolimus and erlotinib. The only condition in which FC101 was not equal or superior in reducing cell viability was when compared to the everolimus and lapatinib combination treatment. In this case, the everolimus and lapatinib combination treatment displayed significantly lower viability than that of treatment with FC101 alone (Figure 6).

### 3.3. Drug Washout Survival Assay

To determine TNBC (MDA-MB-231) cellular growth recovery after drug treatments, cells were treated for 72 h. The growth medium containing the drug was removed and replaced with fresh drug-free growth medium. The cells were placed back in the incubator and allowed to recover for another 72 h. An MTT viability assay was performed on the drug-treated cells at 72 h and on the recovered cells at 72 h. Fold changes in viability levels at 72 h of treatment were graphed against the fold changes in viability levels at 72 h after the removal of drug treatments (Figure 7 and Figure 8).

When the drugs were removed, both lapatinib (1.0 μM) and erlotinib (0.5 μM) single treatment cells were able to recover to the viability level of untreated control cells with no significant difference in viability among the three. All other treatments including FC101 alone and in combination with lapatinib or erlotinib as well as everolimus alone and in combination with lapatinib or erlotinib maintained a significant decrease in viability when compared to untreated cells (*p* < 0.001). However, when comparing the fold change in viability for cells treated for 72 h against the viability fold change of drug washout cells, more marked results were observed. There was no significant difference in viability in cells treated with FC101 (0.5 μM) for 72 h compared to the drug washout cells. Therefore, even when the FC101 treatment was removed, the cells were not able to recover, indicating a robust and durable drug response. This inability to recover was also seen in the cells treated with FC101 (0.5 μM) and lapatinib (1.0 μM) in combination (Figure 7). For all other treatment conditions, the cells recovered their viability levels to a significant extend (*p* < 0.001) extent after the treatments were removed. Cell viability recovered by 66% after lapatinib (1.0 μM) treatment and subsequent drug washout. However, when FC101 (0.5 μM) was added for a combination treatment with lapatinib (1.0 μM), the recovered cell viability was only 11% (Figure 7).

When erlotinib (0.5 μM) was tested after cell treatment and subsequent drug washout, the viability recovered by 74%. In contrast, when FC101 (0.5 μM) was added in combination with erlotinib (0.5 μM), the cell viability recovered by only 23%. FC101 (0.5 μM) treatment alone with drug washout actually provided an additional 8% decrease in viability after the drug was removed. FC101 (0.5 μM) was significantly more effective in sustaining a decrease in viability when compared to everolimus (0.5 μM), because when everolimus was removed, cells recovered by 40%. A 54% recovery of viability was observed when everolimus (0.5 μM) was used in combination with lapatinib (1.0 μM) and then washed out. When everolimus (0.5 μM) was used in combination with erlotinib (0.5 μM) and then washed out, a 20% recovery in viability was observed (Figure 8).

These results demonstrate that FC101 was superior in its ability to provide a sustained decrease in viability both when used alone and when used in combination with lapatinib or erlotinib. Interestingly, when FC101 was used as a single treatment, its viability levels continued to decrease even after its removal, whereas when FC101 was combined with the other drugs, the cells were able to regain some viability.

## 4. Discussion

Cancer drug resistance is a significant clinical challenge that can be effectively addressed by the development of combination therapies [11]. Using both genomic and proteomic information, new combination therapies can be developed that target compensatory oncogenic signaling pathways and achieve better clinical outcomes in treating cancer. Carefully selected combination therapies can be synergistic, blocking both the primary oncogenic target as well as its secondary (drug-induced) targets, which evolve as the cancer cells adapt to survive [11].

TNBC is among the most aggressive types of cancer, and it has the least favorable prognosis [13]. Since TNBC lacks the overexpression of ER, PR, and HER2/Neu, it offers the fewest available therapeutic options. TNBC’s aggressiveness and poor prognosis are both correlated with the overexpression of EGFR [17]. Yet, in clinical trials, patients often do not respond favorably to treatment with EGFR inhibitors alone due to the development of drug resistance [19]. A variety of preclinical and clinical evidence has indicated that combining an EGFR inhibitor with an mTOR inhibitor produces a synergistic drug response that effectively circumvents drug resistance in TNBC [18,19]. There is currently an increasing demand for other novel mTOR inhibitors to establish and validate new drug combinations that can effectively block the oncogenic networks responsible for drug resistance in TNBC. In this context, FC101 also serves as a promising anti-cancer agent for TNBC, because it powerfully modulates the mTOR pathway.

Our original hypothesis was that FC101′s activity against TNBC is similar to that of the known mTOR inhibitor, everolimus, since FC101 reduces the phosphorylation of two key mTOR substrates, S6K and S6 (Appendix A). Since the activity of an EGFR inhibitor (erlotinib or lapatinib) is enhanced when it is combined with an mTOR inhibitor (everolimus), we performed analogous studies with FC101. Phenotypic cellular assays helped assess whether FC101 (in both single and combination treatments) acts similarly to everolimus. The treatment of cultured TNBC cells with FC101 or everolimus along with EGFR inhibitors (erlotinib or lapatinib) was used to measure synergistic effects. The crystal violet and MTT assays revealed a slight additive effect for the combination of FC101 (0.5 μM) and erlotinib (0.5 μM). However, the everolimus combinations generated a more robust synergistic drug response than the FC101 combinations. For example, at 48 h, the combination of FC101 and erlotinib had an additional 14% decrease in viability (compared to FC101 alone). However, everolimus combined with erlotinib had an additional 25% decrease in viability (compared to everolimus alone) (Figure 6).

For combinations involving lapatinib, however, there was a greater difference between FC101 and everolimus. In the MTT assay at 48 h, FC101 (0.5 μM) combined with lapatinib (1.0 μM) had an additional 13% decrease in viability compared to FC101 (0.5 μM) alone, while everolimus (0.5 μM) combined with lapatinib (1.0 μM) showed an additional 52% decrease compared to everolimus alone. Interestingly, while the everolimus/lapatinib (0.5 μM/1.0 μM) combination treatment was more effective at reducing viability than the everolimus/erlotinib (0.5 μM/0.5 μM) combination treatment in the MTT (52% vs. 25%), this difference was not seen with the FC101 combination treatments. The FC101/lapatinib (0.5 μM/1.0 μM) and FC101/erlotinib (0.5 μM/0.5 μM) combination treatments performed similarly (13% vs. 14%).

These findings negate our original hypothesis that FC101 combined with EGFR inhibitors would produce a synergistic effect analogous to that of everolimus in similar combinations. That hypothesis was based on the ability of FC101 to reduce pS6K and p-S6 downstream of mTOR, but it also assumed that FC101 inhibited mTOR directly, which we found not to be true.. This surprising outcome can be explained in part by considering the significant genomic heterogeneity and molecular complexity of cancer. Natural selection ultimately leads to redundancy and crosstalk between cancer cell signaling pathways. For example, similar cancer cell phenotypes can be achieved through a variety of different mechanisms, such as the reduction of p-S6K and p-S6. Notably, our results confirm that FC101′s reduction of p-S6K and p-S6 may not represent direct mTOR inhibition. Thus, FC101 may exploit a unique mechanism that affects mTOR signaling and reduces p-S6K and p-S6. This could include the inhibition of proteins involved in the regulatory crosstalk involving feedback loops between the two canonical EGFR downstream pathways, RAF/MEK/ERK (MAPK) and AKT/mTOR. Extensive crosstalk between these two key pathways is well documented and may be modulated by treatment with FC101. This suggests that a currently unidentified and novel protein interaction may facilitate FC101′s mode of action.

Notably, FC101 as a single treatment outperformed all other single treatments in the crystal violet, MTT, and drug washout assays. For example, in the MTT viability assay (48 h post treatment), the control-normalized drug-induced percentage decrease for FC101, everolimus, lapatinib, and erlotinib treatments was 45%, 30%, 15%, and 10%, respectively. Furthermore, in the drug washout assay, FC101 produced the most robust reduction in viability between the 72 h treatment period and the 72 h recovery period after drug washout. There were no significant differences in the effect of FC101 on cell viability between the two time points, indicating that FC101 was able to affect cells in a sustained manner.

Treatment with FC101 (0.5 μM), alone as well as in combination with lapatinib (1.0 μM) and erlotinib (0.5 μM), produced a sustained decrease in cell viability with no significant difference between the two time points. Moreover, only these three treatment conditions maintained the decrease in cell viability. All other single and combination treatments showed a significant recovery of cell viability between the two time points. FC101 clearly produces a sustained treatment effect, blocking the ability of cancer cells to recover once treatments are removed. This supports the possibility of successful long-term treatment outcomes and reaffirms FC101′s unique and multi-targeting pharmacology.

## 5. Conclusions

Addressing drug resistance using rationally designed combination therapies is an important approach in modern oncology. As a novel anti-cancer agent with unique structure and function, FC101 has significant promise, not only for use as a research tool but also for its clinical applications. In this study, we examined FC101′s mode of action and possible synergy with EGFR inhibitors in the treatment of TNBC. We ruled out our original hypothesis that FC101 combined with EGFR inhibitors would produce a synergistic effect analogous to that of everolimus in similar combinations with EGFR inhibitors. The results demonstrated that FC101 acts through a different mechanism than everolimus as an mTOR signaling modulator. Thus, we have refined our original hypothesis, laying the foundation for a better understanding of FC101′s mode of action. Future studies will include either confirming or ruling out other possible mechanisms responsible for the ability of FC101 to reduce the expression of p-S6K and p-S6, Including protein phosphatase-dependent mechanisms, disruption of protein–protein interactions involved in mTOR signal transduction, and direct or indirect effects upon scaffolding proteins.

## Figures and Tables

**Figure 1 biomedicines-10-02906-f001:**
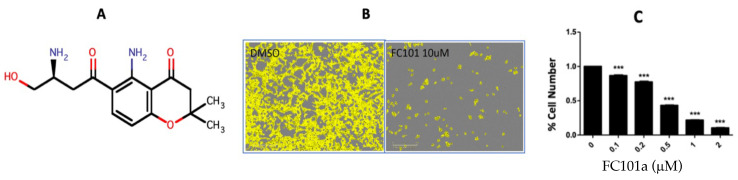
(**A**), the 2D chemical structure of fusarochromanone. (**B**), phase-contrast images of TNBC cells (MDA-MB-231) treated with FC101 (10 μM) and DMSO, 72 h post treatment. (**C**), dose-dependent inhibitory effect of FC101 on TNBC cell growth. *** *p* < 0.01.

**Figure 2 biomedicines-10-02906-f002:**
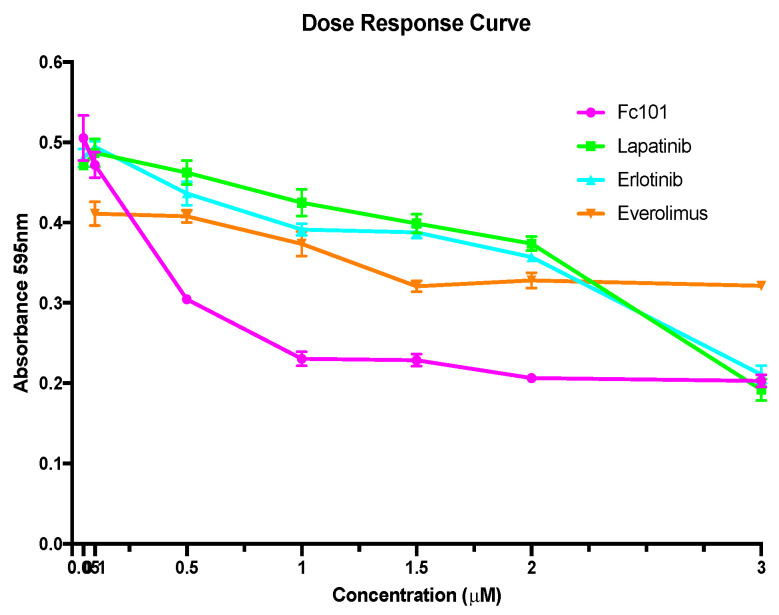
Crystal violet assay dose–response curve, 48 h post drug treatment. The drug panel included FC101, lapatinib, erlotinib, and everolimus at concentrations of 0 μM, 0.05 μM, 0.1 μM, 0.5 μM, 1.0 μM, 1.5 μM, 2 μM, and 3 μM. The graph represents the normalized color intensity A_595_ (% control) vs. the drug concentration.

**Figure 3 biomedicines-10-02906-f003:**
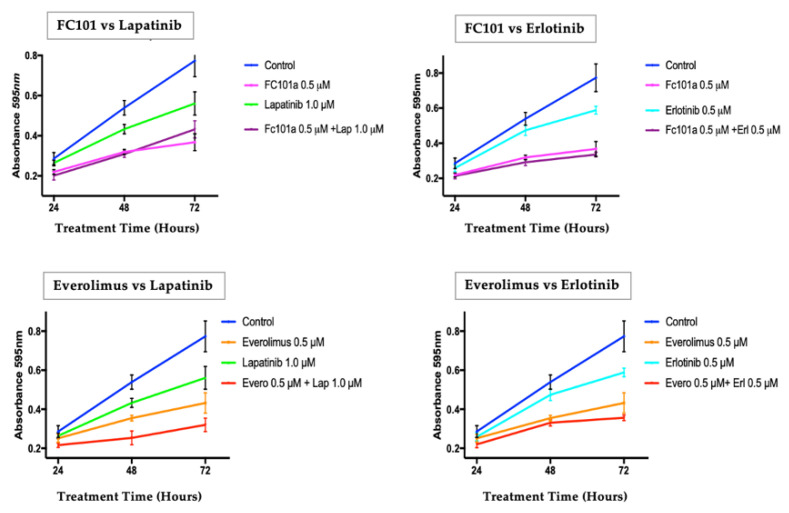
Crystal violet staining assay time response curve. Treatment concentrations included FC101 (0.5 μM), lapatinib (1.0 μM), erlotinib (0.5 μM), everolimus (0.5 μM), and combinations of FC101 or everolimus with lapatinib or erlotinib. The graph represents normalized A_595_ (% control) vs. time course treatments at 24, 48, and 72 h.

**Figure 4 biomedicines-10-02906-f004:**
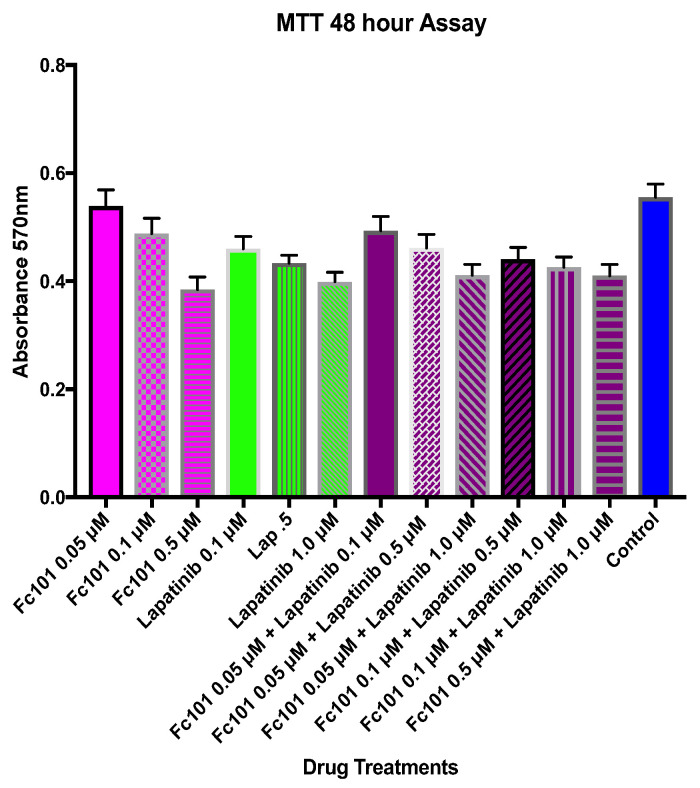
MTT viability assay dose–response curve. FC101 (0.05 μM, 0.1 μM, 0.5 μM) and lapatinib (0.1 μM, 0.5 μM, 1.0 μM). Each concentration of FC101 was paired with an equal or higher concentration of lapatinib. The graph represents the normalized color intensity A_570_ (% control) vs. each single and combination treatment at 48 h.

**Figure 5 biomedicines-10-02906-f005:**
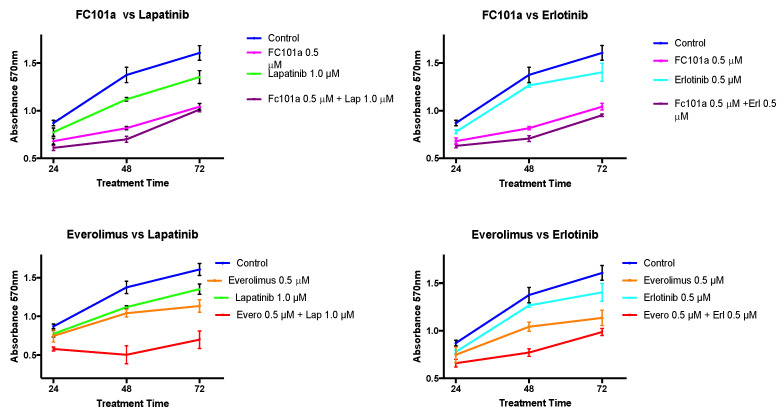
MTT viability assay time response curve. Treatment concentrations included FC101 (0.5 μM), lapatinib (1.0 μM), erlotinib (0.5 μM), everolimus (0.5 μM), and combinations of either FC101 (0.5 μM) or everolimus (0.5 μM) with lapatinib (1 μM) or erlotinib (0.5 μM). The graph represents normalized color intensity A_570_ (% control) vs. time course treatments at 24, 48, and 72 h.

**Figure 6 biomedicines-10-02906-f006:**
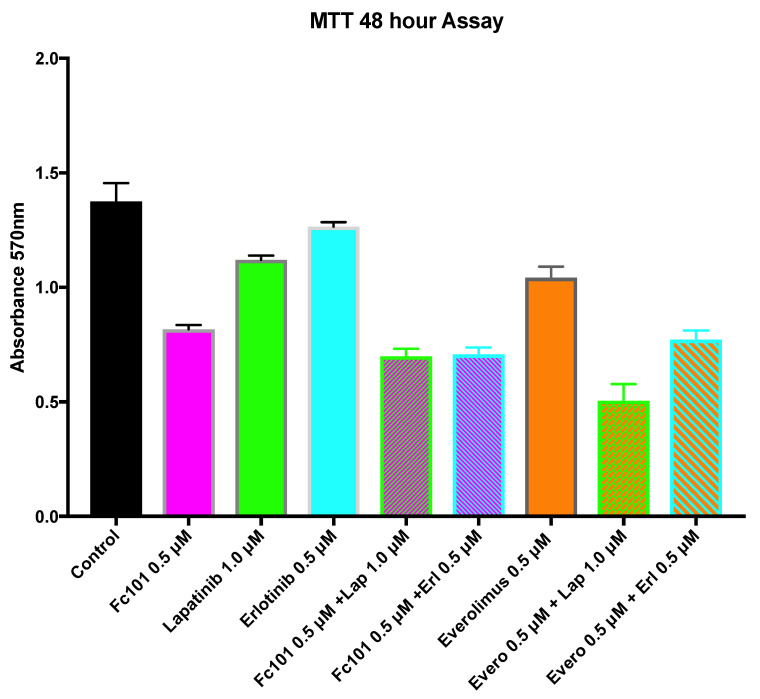
MTT viability assay 48 h post drug treatment. The graph shows the MTT viability assay at the 48 h time point. Treatment concentrations included FC101 (0.5 μM), lapatinib (1.0 μM), erlotinib (0.5 μM), everolimus (0.5 μM), and combinations of FC101 or everolimus with lapatinib or erlotinib.

**Figure 7 biomedicines-10-02906-f007:**
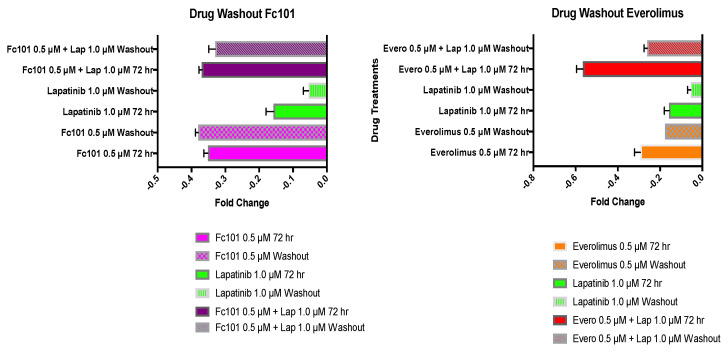
Drug washout assay. Fold changes in viability for the 72 h treatment period are graphed alongside viability fold changes for the 72 h recovery period after drug washout. Solid bars represent the 72 h treatment conditions; patterned bars represent the drug washout conditions. Treatment concentrations included FC101 (0.5 μM), lapatinib (1.0 μM), everolimus (0.5 μM), and combinations of FC101 or everolimus with lapatinib.

**Figure 8 biomedicines-10-02906-f008:**
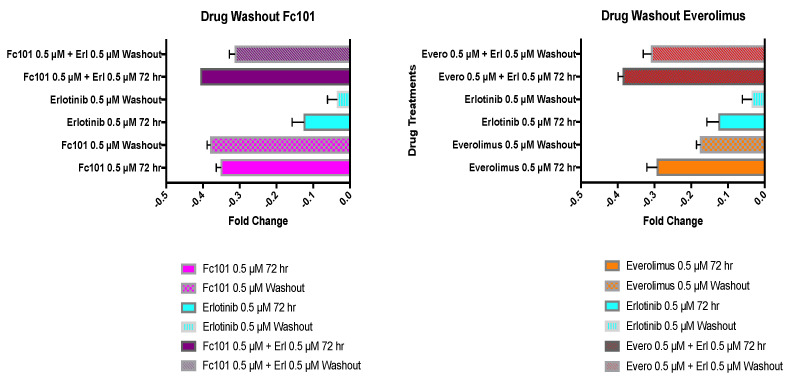
Drug washout assay. Fold changes in viability for the 72 h treatment period are graphed alongside viability fold changes for the 72 h recovery period after drug washout. Solid bars represent 72 h treatment conditions; patterned bars represent the drug washout conditions. Treatment concentrations included FC101 (0.5 μM), erlotinib (0.5 μM), everolimus (0.5 μM), and combinations of FC101 or everolimus with erlotinib.

## Data Availability

The datasets generated and/or analyzed during this study are not publicly available because Natalie Carroll has taken a new job and currently does not have access to the computer that contains the data. However, the datasets are available from the corresponding author upon reasonable request.

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
