# Peer review of "TNBC Therapeutics Based on Combination of Fusarochromanone with EGFR Inhibitors"

_biomedicines, 2022, doi:10.3390/biomedicines10112906_

Round 1
Reviewer 1 Report
Carroll et al. described efficacy of fusarochromanone (FC101) as an anti cancer reagent for the treatment of triple negative breast cancers (TNBC) using MDA-MB231 cell line. In particular, the authors were interested in studying the efficacy of FC101 in terms of its toxicity, anti-proliferative activity and reversibility in comparison with a mTOR inhibitor (everolimus). The initial object was to evaluate FC101’s synergistic activity in combination with two known EGFR inhibitors (erlotinib and lapatinib) through cellular phenotypic screens in TNBC. The results show that FC101 is more potent anticancer reagent respect to the other inhibitors when it is administered alone. However, there is no synergistic effect when it is applied in combination with others. From that point of view, FC101 is inferior to everolimus that has significant synergistic effects in combination with the EGFR inhibitors. Thus, the initial design of the study to evaluate the possibility to use FC101 in combination with other inhibitors is failed. Only advantage to use FC101 seems that the drug response is robust and durable, shown by washout experiments. Some comments on the manuscript are followings:
1. The authors used only one TNBC line, although there are several TNBC lines commercially available. At least 3 of those lines must be used to generalize the concept of the study.
2. Since the mechanism of action of FC-101 has been already studied by others. As mentioned above, if the focus of this work is to evaluate the possibility to use this compound in combination with other EGFR inhibitors, the results are completely negative.
3. Washout experiments do not show any evidence that the compound is more efficacy than other drugs. It might mean that its toxicity could be the problem for normal cells surrounding tumors. Such kind of experiments must be performed in vivo.
4. I found exactly the same results presented by some of the same authors and sent to “Research Square” as preliminary reports available in Internet. I do not think it correct that we reviewers have to work on the manuscript which is already available in public.
From those reasons, I do not consider this manuscript is suitable for publication to Biomedicines.
Author Response
Reviewer 1- Comment 1: The authors used only one TNBC line, although there are several TNBC lines commercially available. At least 3 of those lines must be used to generalize the concept of the study.
To address this concern, we have performed a new set of growth inhibition experiments in three different TNBC cell lines (MDA-MB231, MDA-MB468, BT-549). Our selection criteria for these cell lines are based upon a multi-group analysis of differential proteomics profiles using the iLINCS analytics platform (see Fig S4 heatmap Supplemental Section) which include:
- All three cell types express low protein levels of hormone receptors (ER, PR, IGF) – this explains the lack of targeted treatment options in TNBC.
- All three cell types express high levels of oncogenic proteins, EGFR & ESR1 – this underlines TNBC’s aggressiveness and development of drug resistant.
- All three cell types express low levels of the tumor suppressor protein, PTEN - this underlines poor prognosis in TNBC.
We have measured the inhibition of cultured cell growth of each compound in our drug panel (in both single & combination treatments) in these three cell types to ensure data reproducibility across all these TNBC subtypes. We have used a high-throughput live-cell imaging platform, employing the Incucyte Zoom (ICZ) system for these experiments. We incorporated the growth inhibition graphs from the ICZ experiments (single & combination) in the supplementary section (Fig S5). As shown here, the growth inhibition data for single and combination treatments are consistent across all three cell lines (MB231, MB468, BT-549). This data is also largely consistent with the three independent experiments (crystal violet, MTT, and drug washout screens) on the MB-231 cells that we originally reported in the manuscript.
We reported that FC101 as a single treatment outperformed all the other single treatments in the crystal violet, MTT, and drug washout assays. This data was reproduced in the ICZ experiments across all three TNBC cell lines (see Fig-S5 in the supplemental materials). We had also reported in the manuscript that in the combination treatments, everolimus/lapatinib (0.5 mM/1.0 mM) & everolimus/erlotinib (0.5 mM/0.5mM) were synergistic, while the FC101/lapatinib (0.5 mM/1.0 mM) and FC101/erlotinib (0.5 mM/0.5mM) combination treatment had shown only a slight additive effect. This trend in the differential dual drug responses is also reproduced in the ICZ experiments with respect to the corresponding combination drug treatments in two cell lines, TNBC-231 & TNBC-468 (see Fig S5- Supplemental Section). We noticed disagreement in ICZ data for combination treatments of EVE/LAP and EVE/ERLOT in the TNBC -BT549 cells versus those in TNBC-231 and TNBC-468 cells. We believe that drug precipitation may have occurred in these experiments that interfered with the confluence mask.
Thus, we believe that our data is reproducible and provides robust evidence that negates our original hypothesis that FC101 combined with EGFR inhibitors would produce a synergistic effect analogous to that of everolimus with EGFR inhibitors. Furthermore, the study provides substantive experimental evidence about FC101's MOA to further refine our mechanistic hypotheses and advance our goals for the clinical development of FC101.
Reviewer 1- Comment 2: Since the mechanism of action of FC-101 has been already studied by others. As mentioned above, if the focus of this work is to evaluate the possibility to use this compound in combination with other EGFR inhibitors, the results are completely negative.
We formulated the hypothesis for this study based upon existing knowledge of FC101’s ability to inhibit the oncogenic mTOR pathway. We had experimental evidence from reproducible Western Blotting experiments that showed FC101 reduces p-S6 and pS6K signaling downstream of mTOR. Therefore, we were careful in establishing a connection between existing experimental evidence and our hypothesis about the mechanism of FC101’s mTOR inhibition. While our results negate our original hypothesis that FC101 directly inhibits protein kinases, we demonstrate that FC101’s mTOR signaling activity occurs through an entirely different mechanism than everolimus. This has refined our original hypothesis, and it also lays the foundation for a better understanding of FC101’s mode of action. Since the submission of this manuscript, we also obtained data from a kinome scan screen that rules out FC101’s direct inhibition of kinases. This further supports the results of this study (see Fig. 7 within the supplemental materials). Future studies will also test other possible mechanisms through which FC101 reduces the expression of p-S6 and p-S6K, including protein phosphatase-dependent mechanisms, disruption of protein-protein interactions involved in mTOR signal transduction, and direct or indirect effects upon scaffolding proteins. All of this work provides knowledge that may advance the clinical development of FC101.
Reviewer 1- Comment 3: Washout experiments do not show any evidence that the compound is more efficacious than other drugs. It might mean that its toxicity could be the problem for normal cells surrounding tumors. Such kind of experiments must be performed in vivo.
On the contrary, our results indeed show that FC101 outperforms all of the other single drug treatments in reducing the proliferation and viability of cultured TNBC cancer cells. Additionally, we have shown that FC101 produces the most robust reduction of viability between the 72-hour treatment period and the 72-hour recovery period after drug washout. None of the other drugs (erlotinib, lapatinib, and everolimus) were able to sustain comparable effects upon removal. We plan to explore the underlying mechanism for FC101’s ability to sustain these remarkable anti-proliferation and anti-viability effects in future studies involving both in-vitro and in-vivoexperimental models of cancer.
Reviewer 1- Comment 4: I found exactly the same results presented by some of the same authors and sent to “Research Square” as preliminary reports available in Internet. I do not think it correct that we reviewers have to work on the manuscript which is already available in public.
We initially submitted our manuscript entitled “TNBC Therapeutics Based on Fusarochromanone with EGFR Inhibitors” to BMC Cancer. The pre-print publication on “Research Square” is the result of that submission.
Upon acceptance of publication of this manuscript in Biomedicine, we will request that the “Research Square” pre-print be removed.

Reviewer 2 Report
Caroll et al. describe here the effects of Fusarochromanone on TNBC cell line MDA-MB-231 in a pharmaceutical study. Several other inhibitors are tested in parallel and combined to show efficacy of Fusarochromanone. Although studies like these are always interesting in order to widen the pharmaceutical tool panel for TNBC, this study seems to be unfinished as of yet and would greatly benefit from a more detail-oriented approach and increased care for data management and presentation. The study has merit, but concerns mentioned below need to be thoroughly addressed in order to make it publishable. The general lack of care with manuscript presentation is unusual, at least in this reviewers opinion.
Minor concerns:
· Figures 1 A, B and C should be adequately addressed in the text. The generic Figure 1 in parentheses is not sufficient. It remains also unclear which other publications the graphs/data are taken from. The authors should make very clear who generated displayed data to avoid confusion.
· The statement: “While FC101’s exact mechanism of action is currently unknown, its mode of action may be similar to those of inhibitors of the mTOR and MAPK signaling pathways.“ sounds very confusing…is it known or not? It would be greatly beneficial if the authors could re-write that statement to make their point of view clearer to the reader.
· Since MDA-MB-231 cells were gifted not bought a SNP or STR analysis should be done to verify the cell line.
Major concerns:
· Supplemental Materials mentioned under 1.3 were not accessible for this reviewer.
· A figure 2 seems to be missing.
· The repetition of the methods used and their description in the results section is obsolete and should be removed. Otherwise it seems to artificially inflate the text generated which should be avoided.
· Figure 4 is missing
· Citations are mislabeled (Mahdavian et al. is either 23 or 24 not 22)
Author Response
Review Two - Comment 1: Figures 1 A, B and C should be adequately addressed in the text. The generic Figure 1 in parentheses is not sufficient. It remains also unclear which other publications the graphs/data are taken from. The authors should make very clear who generated displayed data to avoid confusion.
This issue has been addressed. Please see the revised manuscript.
Review Two - Comment 2: The statement: “While FC101’s exact mechanism of action is currently unknown, its mode of action may be similar to those of inhibitors of the mTOR and MAPK signaling pathways.“ sounds very confusing…is it known or not? It would be greatly beneficial if the authors could re-write that statement to make their point of view clearer to the reader.
FC101 is an experimental drug, an attractive candidate for clinical development for two reasons. First, FC101 is a potent cytotoxic agent that affects multiple cancer phenotypes (growth, viability, migration), indicating poly-pharmacology. Second, FC101 is a cancer-specific cytotoxic agent that affects more aggressive cancers more strongly, possibly the result of differential uptake/permeability by aggressive cancers and selective toxicity against them.
Unlike our extensive phenotypic experiments, our mechanistic investigation of FC101's MOA has been limited to a few experiments. Data from Western Blotting screens suggest that FC101 modulates the dual oncogenic mTOR and MAPK signaling pathways downstream of EGFR. Data from a Kinome Scan screen, a competitive binding affinity platform, rules out direct inhibition of kinases as FC101's primary mode of action. Data from cell cycle screens suggest that FC101 is a very strong pro-apoptotic agent that arrests cancer cells in the sub-G1 phase of the cell cycle. However, as yet we have no systematic understanding of the molecular mechanisms underpinning FC101's induced drug response in cancer. Identifying its molecular MOA is significant because it can reveal critical therapeutic targets for clinical development.
Review Two -Comment 3: Since MDA-MB-231 cells were gifted not bought a SNP or STR analysis should be done to verify the cell line.
The cell lines were purchased from ATCC by INLET-LSUHSC and used for this project at the same research facility. Upon initial receipt, cell lines were expanded and frozen at low passage number to guarantee that all cells are brought into culture are at low passage number and prevent accumulation of unknown passage-related changes. Standard laboratory protocols were also in place to maintain consistency with regard to number of cells passed, time to passage, and passage procedure.
Review Two -Comment 4: Supplemental Materials mentioned under 1.3 were not accessible for this reviewer.
This issue has been addressed. Please see the supplemental section in the updated submission.
Review Two -Comment 5:
A figure 2 seems to be missing.
This issue has been addressed. Please see the revised manuscript.
Review Two - Comment 6: The repetition of the methods used and their description in the results section is obsolete and should be removed. Otherwise it seems to artificially inflate the text generated which should be avoided.
This issue has been addressed. Please see the revised manuscript.
Review Two -Comment 7: Figure 4 is missing
This issue has been addressed. Please see the revised manuscript.
Review Two -8: Citations are mislabeled (Mahdavian et al. is either 23 or 24 not 22)
Mahdavian et al. synthesized the experimental drug, FC101, the synthesis paper has not yet been published. Please see the revised manuscript. Citations #23 & #24 describe phenotypic screens for anti-cancer properties of FC101.
Round 2
Reviewer 1 Report
All the comments raised by the reviewer are well taken and some of the results obtained form additional experimental works are presented in the Supplementary Materials. Thus the manuscript is now acceptable for publication.
Author Response
Thank you for acknowledgement.
Reviewer 2 Report
Although the authors tried to address most of the criticisms, but there are still two minor issues:
- supplemental files are generically addressed in the text as supplemental material yet the files are labeled S1-S5. The references should be accurate in the text (label them as S1, S2, S3 and so on when citing the data).
- it is still common practice these days to authenticate a cell line, so there is no reason not to do it, even when the authors describe the origin of the cells.
Once the cell line is authenticated (upload STR/SNP analysis as supplemental material) and the supplemental material is adequately referenced, the manuscript is acceptable for publication.
Author Response
We have addressed the two requested minor revisions.